# *HelioX*: A GPU-Native Framework for Simulation and Training of Biophysically Detailed Networks

**Junfeng Lu** [1]  **Zijie Yu** [2]  **Shaoyang Cui** [3]  **Gan He** [3]  **Ruiqin Xiong** [1]  **Kai Du** [3]  **Tiejun Huang** [1]

## Abstract

Biophysically detailed neural networks represent a promising frontier for brain-inspired AI, offering intrinsic spatio-temporal dynamics to enhance the expressivity and computational density of deep learning systems. However, general-purpose deep learning frameworks suffer from a fundamental mismatch between their dense parallel optimizations and the irregular, tree-structured complexity of biological mechanisms. In this work, we propose **HelioX**, a **GPU-native** framework designed to unify high-performance simulation with scalable training. Unlike approaches that adapt biology to existing deep learning tools, **HelioX** adopts a "GPU-to-Biophysics" paradigm. We tailor the underlying GPU parallelism to biological structures by implementing custom-fused CUDA kernels for both the Dendritic Hierarchical Scheduling (DHS) algorithm and its gradient propagation. This design eliminates the runtime overhead of generic automatic differentiation and enables multi-stream concurrency for spike generation and equation assembly. Experimental results demonstrate that **HelioX** outperforms standard simulators (NEURON) by orders of magnitude and surpasses prior GPU-based solvers in both speed and scalability. We successfully train deep biophysical MLPs and whole-brain-scale biophysical circuits (e.g., the BAAIWorm *C. elegans* model) on a single consumer-grade GPU. **HelioX** establishes a new standard for computational efficiency, enabling the training of biophysically detailed models at scales previously unattainable.

[1]School of Computer Science, Peking University, Beijing, China [2]School of Electronics Engineering and Computer Science, Peking University, Beijing, China [3]Department of Psychological and Cognitive Sciences, Tsinghua University, Beijing, China. Correspondence to: Kai Du <kai_du@tsinghua.edu.cn>, Tiejun Huang <tjhuang@pku.edu.cn>.

*Proceedings of the 43rd International Conference on Machine Learning*, Seoul, South Korea. PMLR 306, 2026. Copyright 2026 by the author(s).

## 1. Introduction

The remarkable prosperity of modern artificial intelligence is rooted in a foundational act of elegant simplification: the point-neuron abstraction. By distilling the intricate complexity of biological brains into scalar integration and non-linear activations, this paradigm catalyzed the era of big data and large-scale model training. However, this abstraction discards a fundamental property of biological intelligence: spatiotemporal dynamics. Recent advances like Neural ODEs (Chen et al., 2018) and Liquid Time-Constant Networks (Hasani et al., 2021) demonstrate that reintroducing continuous-time dynamics yields superior robustness and data efficiency, suggesting that shifting from "instantaneous" inference to dynamic systems is a key frontier for AI. A natural next step is to incorporate not only time but also spatial structure into the computational unit.

If these models represent the first steps toward dynamic AI, biophysically detailed multi-compartment models (hereafter referred to as **detailed neurons**) represent its logical zenith. Unlike point-like dynamic systems, detailed neurons reintroduce spatial morphology as a primary computational dimension. Evidence suggests that this structural complexity is not mere biological redundancy but a source of immense power: a single detailed neuron can function as a 5-to-8 layer deep neural network (Beniaguev et al., 2021), natively executing sophisticated operations such as XOR logic and coincidence detection through the intrinsic interplay of its dendritic branches (Gidon et al., 2020). By coupling non-linear dynamics with physical morphology, detailed neurons achieve a level of computational density that standard paradigms cannot replicate, positioning them as the canonical candidate for next-generation, high-expressivity AI.

However, the transition from point-neurons to detailed biophysical models is bottlenecked by the lack of a scalable training framework. Unlike the forward pass of standard ANNs, which benefits from highly parallelizable matrix operations, detailed neuron models' forward pass is a fine-grained time-stepping simulation process that repeatedly solves coupled ODEs on irregular dendritic trees. As a result, end-to-end training, which couples simulation, objective evaluation, gradient propagation, and parameter up-

dates, becomes both time-consuming and memory-intensive. Existing high-performance simulators designed for GPU parallelism often remain non-differentiable, while modern differentiable frameworks struggle with the topological sparsity of dendritic trees. Because these frameworks wrap biophysical equations into general-purpose engines designed for regular ANNs, they suffer from prohibitive memory overhead and computational redundancy, rendering the training of large-scale, morphologically detailed networks intractable(Deistler et al., 2025).

In response to these challenges, we develop *HelioX*, a GPU-native engine specifically architected to unify high-performance simulation with efficient differentiable training for biophysically detailed networks (BDNs). Importantly, *HelioX* is designed to complement NEURON's modeling ecosystem. Models are built in NEURON using existing workflows, while *HelioX* serves as a NEURON-compatible, GPU-native backend that accelerates simulation and enables efficient differentiable training.

*HelioX* integrates a simulation core and a gradient computation unit within a cohesive, high-throughput framework. On the simulation side, *HelioX* provides a native implementation of the DHS, further optimized through operator reduction and stream concurrency to maximize hardware utilization and memory efficiency. The training module features a dedicated gradient module that directly interfaces with the simulation core to track state variables. This design enables precise backpropagation across diverse neuronal morphologies, leveraging optimized gradient algorithms ((Zhang et al., 2023; Zhao et al., 2024)) to support large-scale network training. To facilitate accessibility, *HelioX* offers a high-level modular API, allowing researchers to construct multi-layer biophysical architectures with ease. Our contributions are summarized as follows:

**(a) GPU Native Simulation Engine:** We develop a GPU-native engine based on the DHS method. By implementing this theoretically optimal parallelization strategy through custom CUDA kernels, we maximize hardware utilization and achieve unprecedented simulation speed for complex dendritic structures.

**(b) Effective Training Module:** We implement an efficient training module that provides GPU-native support for specialized gradient algorithms of detailed neurons. This framework includes a modular Python API for constructing biophysical MLPs, facilitating the scalable training of large-scale biophysical circuits.

**(c) System Validation and Benchmarking:** We demonstrate the efficacy of *HelioX* across diverse biophysical learning tasks. On the MNIST(LeCun et al., 2002) benchmark, we successfully train 3- to 5-layer biophysical MLPs that maintain stable accuracy while delivering significant

gains in speed and memory efficiency. Furthermore, on the BAAIWorm *C. elegans* circuit-fitting task(Zhao et al., 2024), *HelioX* substantially accelerates the optimization loop and reduces peak GPU memory from 47GB to ~5GB, enabling whole-circuit training on a single consumer GPU.

**Why existing approaches fail.** Current systems struggle to make *training* of large morphologically detailed networks practical because several bottlenecks compound in the inner loop:

- **Irregular tree solves do not map to dense-tensor kernels:** the cable equation induces a tree-structured sparse solve whose dependency pattern breaks standard GEMM-centric optimizations.

- **High overhead from fine-grained mechanism execution:** heterogeneous mechanisms lead to many small GPU kernels and frequent launch/synchronization overhead, especially when within-step ordering must be preserved.

- **Host-mediated event handling causes synchronization:** sparse spike delivery is latency-sensitive; naive CPU involvement introduces repeated host–device round trips and stalls.

- **AD-based training is memory/compute heavy:** differentiating through long-horizon ODE integration creates large computation graphs and state tapes, quickly becoming prohibitive for multi-compartment networks.

**Conflict of Interest Disclosure.** The authors declare no financial conflicts of interest related to this work beyond the research funding disclosed in the Acknowledgements.

## 2. Background

**From inference to simulation.** Unlike standard artificial neural networks where *inference* is a sequence of feed-forward matrix operations, biophysically detailed neural networks (BDNs) are governed by continuous-time biophysical dynamics. Given time-varying inputs and initial states, the forward pass of a BDN is therefore a *numerical simulation* process: it advances the neuronal state over time according to biophysical differential equations. In this paper, we use **simulation** to denote the forward computation of a BDN.

**Single and multi-compartment dynamics.** BDNs are governed by continuous-time biophysical dynamics. After spatial discretization, a neuron is represented as a tree of coupled compartments. Let $V_i(t)$ be the membrane potential of compartment $i$ with capacitance $C_i$. A compact form of

the discretized cable equation is

$$C_i \frac{dV_i(t)}{dt} + I_{\text{mem},i}(t) = I_{\text{ext},i}(t) + \sum_{j \in \mathcal{N}(i)} I_{i \leftarrow j}(t), \tag{1}$$

where $\mathcal{N}(i)$ denotes axially connected neighbors and $g_{ij} \triangleq 1/R_{ij}$ is the axial conductance. We denote the axial current from compartment $j$ to $i$ as $I_{i \leftarrow j}(t) \triangleq g_{ij}\left(V_j(t) - V_i(t)\right)$. The transmembrane current aggregates biophysical mechanisms,

$$\begin{aligned} I_{\text{mem},i}(t) \triangleq\ & I_{\text{ion},i}\left(V_i(t), \mathbf{g}_i(t); \boldsymbol{\theta}_{\text{ion}}\right) + I_{\text{syn},i}(t) \\ & + I_{\text{leak},i}\left(V_i(t)\right), \end{aligned} \tag{2}$$

and gating states typically follow additional ODEs,

$$\frac{d\mathbf{g}_i(t)}{dt} = \Phi\left(\mathbf{g}_i(t), V_i(t); \boldsymbol{\theta}_{\text{ion}}\right). \tag{3}$$

**Simulation pipeline.** In a biophysically detailed simulator, the forward computation is a fixed-step time integration that advances compartment voltages and mechanism states. We implement the model as a collection of **Mech** modules (e.g., ionic channel kinetics, synaptic dynamics, passive leak, and externally injected currents such as `IClamp`), each contributing transmembrane currents and/or local state updates.

A single simulation step can be viewed as a structured pipeline: (i) *event-driven network coupling*, which propagates spikes and applies synaptic effects (with delays) to update synaptic states; (ii) *mechanism evaluation*, which updates local states and accumulates transmembrane currents (including the synaptic currents determined by the delivered events); and (iii) *cable equation assembly and solve*, which advances voltages by solving the tree-structured coupling induced by cable theory.

**Challenges for GPU-native simulation.** This pipeline poses several challenges for GPU-native simulation: (1) the cable equation yields an irregular tree-structured solve that is poorly matched to dense-tensor optimizations; (2) heterogeneous mechanisms interact through shared states, requiring strict within-step ordering and explicit dependency enforcement under multi-stream execution; and (3) spike propagation is sparse yet latency-sensitive, and naive host-mediated handling can introduce frequent host-device synchronization.

**Learning beyond automatic differentiation.** Automatic differentiation (AD) enables end-to-end training of dynamical models by differentiating through the numerical solver. Recent work such as JAXLEY demonstrates that AD-based pipelines can support learning in multi-compartment neurons. However, backpropagating through the full ODE

integration procedure can incur substantial computational and memory overhead due to long time horizons and large state spaces. An alternative direction is to derive analytical, structure-aware gradients that exploit the biophysical model structure.

DeepDendrite ([Zhang et al., 2023](#)) proposes an analytical synaptic gradient for training BDNs. A common choice of forward readout is a time-averaged somatic voltage over a learning window,

$$\bar{v}_j \triangleq \frac{1}{t_e - t_s} \int_{t_s}^{t_e} v_j(t)\, dt, \tag{4}$$

from which task-specific predictions are computed and a loss $E$ is evaluated. The key quantity for learning is the local error signal (gradient)

$$\delta_j \triangleq \frac{\partial E}{\partial \bar{v}_j}, \tag{5}$$

which can be computed at the output layer and propagated to hidden layers. Given $\delta_j$, the per-timestep synaptic update for the $k$-th synapse from neuron $i$ to neuron $j$ is

$$\Delta W_{ijk}^{(n)} = \delta_j\, r_{ijk}\, g_{ijk}\, f\left(v_i(t_n)\right), \quad t_n \triangleq t_s + n\, dt, \tag{6}$$

where $g_{ijk}$ is the synaptic conductance and $r_{ijk}$ is the transfer resistance from the dendritic compartment hosting synapse $(i \to j, k)$ to the soma of neuron $j$. This morphology-dependent $r_{ijk}$ makes the rule applicable to general multi-compartment neurons; a single-compartment neuron is a special case where $r_{ijk}$ reduces to the compartment input resistance. Finally, weights are updated by accumulating $\Delta W_{ijk}^{(n)}$ over a learning window,

$$W_{ijk} \leftarrow W_{ijk} - \eta\, \frac{dt}{t_e - t_s} \sum_{n=0}^{N-1} \Delta W_{ijk}^{(n)}, \quad N \triangleq \frac{t_e - t_s}{dt}. \tag{7}$$

Transfer and input resistances can be computed by a reference simulator (e.g., NEURON).

**Challenges for efficient learning systems.** While analytical gradients provide an efficient learning rule, realizing high training throughput requires careful system design: (1) neuronal morphologies are heterogeneous and their states are distributed sparsely across dendritic trees rather than stored in regular dense tensors, often leading to fragmented, fine-grained operators and low GPU utilization; (2) different learning rules and neuron types require accessing biophysical quantities at different locations (e.g., compartment voltages, synaptic conductances, transfer resistances), demanding flexible indexing and efficient gathering/scattering of irregular data; and (3) training interleaves forward simulation with gradient computation and parameter updates, which requires low-overhead data exchange and synchronization between the simulation core and the learning modules.

## 3. Related Work

### 3.1. Biophysically Detailed Models

While point neurons abstract cells into single nodes, detailed neurons explicitly incorporate morphology as a primary computational dimension. Grounded in Cable Theory (Rall, 1959), the dendritic tree is discretized into hierarchical compartments where membrane potentials evolve via coupled systems of ODEs. These systems are driven by transmembrane ion channel kinetics (Hodgkin & Huxley, 1952). This morphological complexity translates into powerful structural inductive biases. Rather than acting as passive cables, dendrites function as active nonlinear processors (London & Häusser, 2005) that provide a biophysical substrate for high dimensional computation. Mathematically, this structure has been linked to deep network expressivity (Beniaguev et al., 2021). This enables individual neurons to natively execute complex operations such as XOR logic and direction selective filtering (Gidon et al., 2020), functions that would otherwise require multilayer ANNs.

### 3.2. Simulation and Training Frameworks

Although traditional simulators such as NEURON and CoreNEURON (Hines, 1984) remain the gold standard for accuracy, their lack of native automatic differentiation makes them incompatible with modern gradient-based pipelines. DeepDendrite (Zhang et al., 2023) introduced the Dendritic Hierarchical Scheduling (DHS, (Zhang et al., 2023)), a theoretically optimal method, to maximize GPU parallelism for simulation, yet it lacks a differentiable wrapper, leaving a gap between high-performance simulation and end-to-end training. Learning frameworks such as Spiking-Jelly(Fang et al., 2023) effectively integrate spiking neural networks with PyTorch but are tailored for simple leaky integrate and fire models. They do not support the hierarchical compartment topologies or complex systems of equations required for detailed morphological modeling. Existing differentiable biophysical frameworks such as BrainPy (Wang et al., 2023) and JAXLEY (Deistler et al., 2025) represent the state of the art by leveraging automatic differentiation engines like JAX. These platforms adopt a generic mapping approach that translates biophysical equations into standard computational primitives. Without domain specific sparse solvers, the computation graph for dendritic trees becomes prohibitively large during backpropagation. This leads to severe memory overhead and limits scalability to small circuits.

## 4. The *HelioX* Framework

### 4.1. GPU-Native Simulation Framework

***HelioX*** is *GPU-native by design*: we treat the GPU as the primary execution backend rather than an optional accelerator. Accordingly, we co-design data layouts, kernel boundaries, and scheduling around GPU constraints, and implement a set of targeted optimizations that map the simulator's irregular structure onto efficient GPU execution.

Concretely, we organize our GPU-native simulation stack into four components that correspond to the optimizations discussed below: (1) **Spike Processing** for sparse event handling, (2) **Unified Mechanism Template and Variable Registry** (MechTemp/VarStruct) for consistent mechanism instantiation and stable state access, (3) **Concurrent ODE Construction** for multi-stream mechanism evaluation and assembly, and (4) **Parallel ODE Solving** for the tree-structured cable solve.

**Spike Processing.** Biophysical events (spikes) are characterized by high temporal sparsity, often occurring in less than 1% of the simulation steps. Traditional simulators suffer from frequent host-device synchronizations to detect these rare events. To exploit this, we implement a two-stage event pipeline that replaces unconditional data transfers with a 4-byte gating mechanism. In each timestep, a lightweight kernel performs spike detection and returns only a single integer (spike count) to the host via pinned memory. If the count is zero, the framework bypasses all subsequent heavy operations, including the back-copy of spike indices and the invocation of post-synaptic delivery kernels. This design reduces the per-step overhead to $O(1)$ on spike-free steps, effectively eliminating redundant PCIe traffic and control-flow overhead in sparse networks.

**Unified mechanism template and variable registry.** *HelioX* introduces a template-based mechanism abstraction (MechTemp) to avoid backend divergence across CPU and GPU. In contrast to systems where CPU and GPU acceleration paths evolve with separate backends and code-generation toolchains (e.g., NEURON vs. CoreNEURON), each biophysical mechanism in *HelioX* is specified once as a node-level update and automatically instantiated into CPU and GPU kernels with consistent within-step semantics (differences are limited to floating-point rounding).

Mechanism parameters and state variables are stored in a unified structured layout (VarStruct) and accessed through a uniform `var(var_name)` interface, enabling consistent addressing and memory interpretation across CPU/GPU.

In contrast, JAX/XLA-based frameworks (e.g., JAXLEY) abstract away device buffers and may reorder/reshape memory via compiler passes, which complicates enforcing a

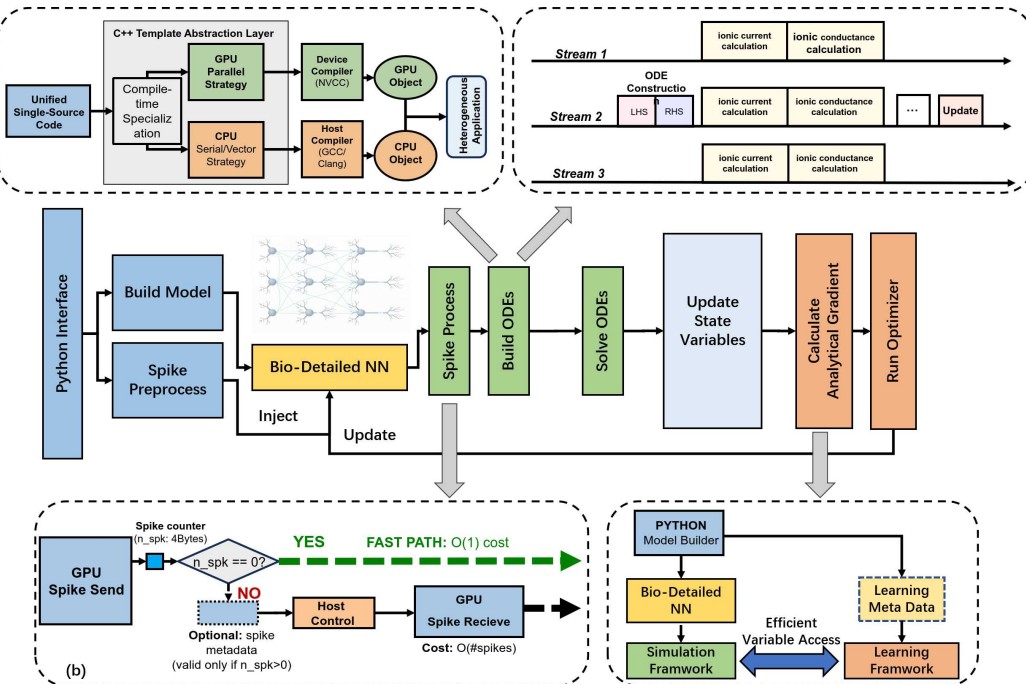

*Figure 1.* **Overview of *HelioX*. *HelioX*** bridges NEURON model specification and GPU-native execution. The runtime integrates (1) sparse spike processing, (2) a unified mechanism template and variable registry (MechTemp/VarStruct), (3) multi-stream concurrent ODE construction, (4) DHS-based parallel tree solving for the cable equation, and (5) an analytical-gradient learning module that directly binds to simulator states for efficient training.

fixed layout and directly binding learning code to simulator states.

***HelioX*** therefore exposes a reflective *variable registry* with stable identifiers that let external learning code (e.g., Python) locate and read/write simulator variables without ad-hoc name-to-index glue. This registry acts as a lightweight ABI between simulation and learning, improving interoperability and scalability.

**Concurrent ODE construction.** We decompose each fixed-step timestep into a set of GPU tasks (e.g., spike delivery, mechanism current evaluation, RHS/diagonal assembly, mechanism state updates, and recording/`VecPlay`) with explicit data dependencies . We exploit **multi-stream concurrency** to run independent mechanism kernels in parallel, and enforce NEURON-equivalent within-step semantics using **CUDA events** (`cudaStreamWaitEvent`) instead of device-wide synchronization. This event-driven scheduling reduces CPU synchronization overhead and improves GPU utilization when mechanism kernels are small and heterogeneous. To further reduce GPU kernel launch overhead, we apply lightweight kernel fusion to cut launch overhead and redundant memory traffic.

**Parallel ODE Solving.** Solving the multi-compartment cable equation reduces to a sparse linear solve on a tree, typically performed by the Hines method. This step is challenging to parallelize because the elimination and back-substitution must follow a strict dependency order induced by the dendritic tree; naive parallel execution can violate these dependencies and thus produce incorrect results.

To address this, we adopt the **Dendritic Hierarchical Scheduling (DHS)** algorithm from DeepDendrite (Zhang et al., 2023). DHS organizes the tree solve into dependency-respecting levels, exposing maximal parallelism while preserving the exact Hines semantics. DeepDendrite proves DHS to be theoretically optimal in terms of parallel time for fine-grained neuron (multi-compartment) Hines solves, making it a principled foundation for GPU-native ODE solving.

### 4.2. Learning Framework

***HelioX*** adopts an *analytical-gradient* learning approach rather than relying on generic automatic differentiation (AD).

By leveraging structure-aware gradient derivations for biophysical models, analytical gradients avoid mechanically backpropagating through long-horizon ODE integration, and thus substantially reduce both compute and memory overhead.

This advantage is critical for morphologically detailed neu-

rons: a single cell may contain hundreds of compartments and heterogeneous mechanisms, and a network forward pass typically corresponds to a continuous simulation trajectory (e.g., tens to thousands of milliseconds) discretized into many timesteps. As a result, training produces large volumes of irregular, time-indexed biophysical signals (voltages, conductances, and mechanism states), making efficient *binding, indexing, aggregation, and update* of such data a primary systems challenge.

**Modeling–execution workflow and freeze point.** *HelioX* is designed to be compatible with the established NEURON modeling ecosystem. Model specification—including morphology parsing, mechanism insertion, connectivity construction, and event routing—is performed in NEURON.

After model construction, *HelioX* imports the model into its GPU-native runtime and enters an *execution phase* in which all learnable parameters and training signals are bound to a stable set of runtime identifiers. This *freeze point* decouples high-frequency training from frontend object semantics: the learning backend only consumes a compact metadata description (e.g., the set of trainable variables, optimizer configuration, and hyperparameters) and operates directly on the simulator's device-resident states.

**Handle-based state binding and temporal buffers.** To support learning on irregular dendritic trees, *HelioX* exposes a reflective variable registry that maps biophysical quantities to their underlying GPU buffers.

However, repeatedly resolving variables by name or metadata at every timestep would introduce non-negligible overhead in the tight training loop. Therefore, *HelioX* uses a *handle-based caching* mechanism: variable resolution is performed once and cached as stable handles (device pointers plus shape/stride metadata), enabling $O(1)$ access for subsequent reads/writes during training.

Different objectives require different temporal views of simulator states. For time-averaged readouts (e.g., voltage averages over a learning window), *HelioX* supports *in-simulation accumulation* so that reductions are performed online during simulation, reducing post-processing and data movement. For trajectory-level objectives that require time series, the runtime maintains a device-side temporal buffer (window/ring buffer) with contiguous layout, allowing learning kernels to consume sequential data efficiently without repeated fragmented gathering.

**Two learning workflows: structured composition and general biophysical models.** *HelioX* provides two complementary workflows to cover common research settings.

For structured feed-forward architectures (e.g., biophysical MLPs built from simplified or passive neuron models),

*HelioX* offers a PyTorch-like `Sequential` API for composing multi-layer networks and a batched input-encoding interface (e.g., converting a vector input into rate-coded spike sequences—one per input dimension, to drive input neurons) to reduce Python-side overhead. For general biophysical circuits with heterogeneous morphologies, mechanisms, and recurrent connectivity, users build the model in NEURON and then invoke *HelioX*'s learning runtime by passing the exported model along with the training configuration; the runtime binds trainable parameters via the variable registry and executes simulation, gradient computation, and parameter updates on the GPU.

## 5. Experimental Results

### 5.1. Numerical Fidelity and Scalability

We evaluate *HelioX* by comparing numerical fidelity and runtime efficiency against established simulators (NEURON, CoreNEURON, and DeepDendrite). All experiments run on an AMD Ryzen 9 9950X and an NVIDIA RTX 5090 (32GB), spanning a single detailed L5 pyramidal cell (L5PC) up to a 500-cell passive HPC network; *HelioX* uses FP64 throughout to match NEURON.

We measure accuracy by the maximum absolute error (MAE) of membrane potentials and spike-time consistency, using NEURON as reference. Figure 2 shows MAE below $10^{-8}$ mV (i.e., at floating-point round-off level), which is far smaller than voltage changes induced by typical integration step sizes; accordingly, in all our tests *HelioX* reproduces identical spike times as NEURON. Additional fidelity results for the passive 500-HPC benchmark are reported in Appendix A.

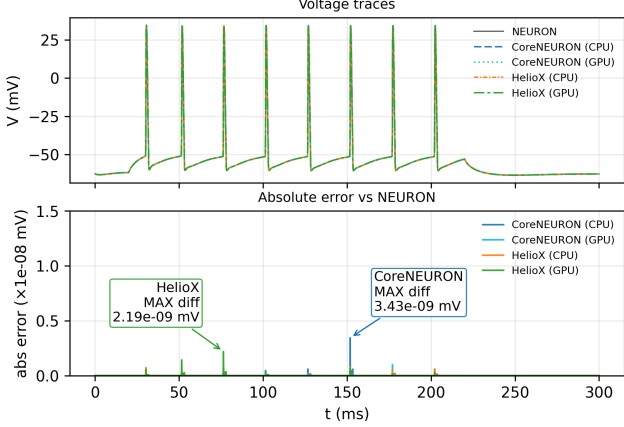

*Figure 2.* **Numerical fidelity of *HelioX* (L5PC).** We compare *HelioX* against NEURON on a biologically detailed Layer 5 pyramidal cell (L5PC). *HelioX* matches NEURON with maximum absolute error below $10^{-8}$ mV and reproduces identical spike timing.

*Table 1.* Comprehensive performance comparison across simulation benchmarks. *HelioX* achieves up to $\sim 2.25\times$ speedup compared to the strongest baseline (CoreNEURON). The best result in each row is highlighted in **bold**. The speedup factor (in parentheses) is calculated relative to CoreNEURON. "-" indicates the method does not support the task or data is unavailable.

| Benchmark Task | NEURON | DeepDendrite | JAXLEY | CoreNEURON | *HelioX* |
|---|---|---|---|---|---|
| Single L5PC | 2.45 | 3.30 | - | 2.99 | **1.33 (2.2$\times$)** |
| L5PC Network (20 cells) | 48.87 | 4.05 | - | 2.82 | **1.56 (1.8$\times$)** |
| Passive 500 HPC | 131.48 | 5.61 | 10.30 | 5.34 | **3.30 (1.6$\times$)** |
| Active 500 HPC | 134.17 | 9.55 | - | 8.04 | **3.92 (2.1$\times$)** |

Having established this high degree of numerical correctness, we further examine the computational advantages of our framework. Computational efficiency is measured by the speedup ratio in wall-clock execution time relative to the baselines. Through our experiments over the models and frameworks mentioned above, we find that *HelioX* reaches a speedup of 1.61–2.25$\times$ over CoreNEURON and 1.69–2.60$\times$ over DeepDendrite, as summarized in Table 1. For JAXLEY, we take a conservative reporting strategy. We include only the Passive 500-HPC case, the simplest setting for which we could construct the closest NEURON-aligned counterpart. Even in this setting, the reproduced traces showed observable discrepancies from the NEURON reference (0.0817 mV without spikes), and the mismatch became substantially larger when delayed spike-driven interactions were involved (up to 12.89 mV). Since timings under mismatched numerical semantics may conflate simulator efficiency with model-definition differences, we omit the more complex JAXLEY entries rather than report non-comparable measurements. On this JAXLEY-comparable Passive 500-HPC setting, *HelioX* achieves a significant speedup of 3.12$\times$. These results demonstrate that *HelioX* effectively combines the numerical fidelity of traditional CPU simulators with the superior throughput of modern GPU hardware.

### 5.2. Large-Scale Biophysical Network Training: *C. elegans* Whole-Brain Simulation

We use the BAAIWorm whole-circuit fitting task as our primary learning validation. This is not a toy problem, but an whole-brain-scale, recurrent, biophysically detailed optimization task that was previously extremely demanding in both compute and memory; our goal here is to substantially lower that practical barrier.

Concretely, we replicate the biological data fitting task from BAAIWorm, a recently proposed whole-brain model of *C. elegans* (Zhao et al., 2024). The benchmark optimizes network parameters of a biologically detailed circuit so that its simulated activity matches experimental recordings.

The target circuit comprises 136 multi-compartment Hodgkin–Huxley neurons (sensory neurons, interneurons, and motor neurons) connected according to the *C. elegans*

*Table 2.* Training time (per epoch) and peak GPU memory on the BAAIWorm *C. elegans* circuit-fitting task (RTX 4090). Speedup is reported in parentheses, and memory savings are reported as percentage reduction ($\downarrow$).

| Method | Time / epoch (s) | Peak GPU mem. (GB) |
|---|---|---|
| CoreNEURON | 14840 | 47.4 |
| *HelioX* | 5100 (2.91$\times$) | 5.2 ($\downarrow$ 89.0%) |
| *HelioX* (DS=5) | 60 (247.33$\times$) | 1.6 ($\downarrow$ 96.6%) |

connectome (Zhao et al., 2024). Following BAAIWorm, we treat both chemical synapses and electrical gap junctions as trainable and optimize their strengths; the original work also considers additional trainable factors such as synapse polarity (excitatory vs. inhibitory) and external input currents to a subset of sensory neurons (Zhao et al., 2024).

**Training target and loss.** BAAIWorm defines the training target using whole-brain calcium imaging. Specifically, it computes the Pearson correlation matrix from experimentally recorded $Ca^{2+}$ signals of 65 identified head neurons. We minimize the mean squared error (MSE) between the correlation matrix produced by the simulation and the experimental correlation matrix (Zhao et al., 2024).

**Optimization algorithm.** Following BAAIWorm, we adopt the optimization algorithm in (Zhao et al., 2024) for multi-compartment recurrent circuits. At a high level, it performs gradient-based optimization and uses mathematical approximations to compute *approximate gradients* for the recurrent network, which are then used to update trainable synaptic and coupling parameters. We adopt the same task definition and validate *HelioX* by matching the training trajectory against a CoreNEURON-based baseline.

**Learning-time downsampling.** Detailed-neuron simulation typically requires a fine integration step (e.g., 0.5 ms), while many learning signals evolve smoothly at this temporal resolution. *HelioX* therefore supports *learning-time downsampling* for gradient computation: we compute gradients using a temporally downsampled subset of the simulated trajectory (while keeping the forward simulation at the original step size). This reduces the effective history length and the number of gradient evaluations, lowering compute

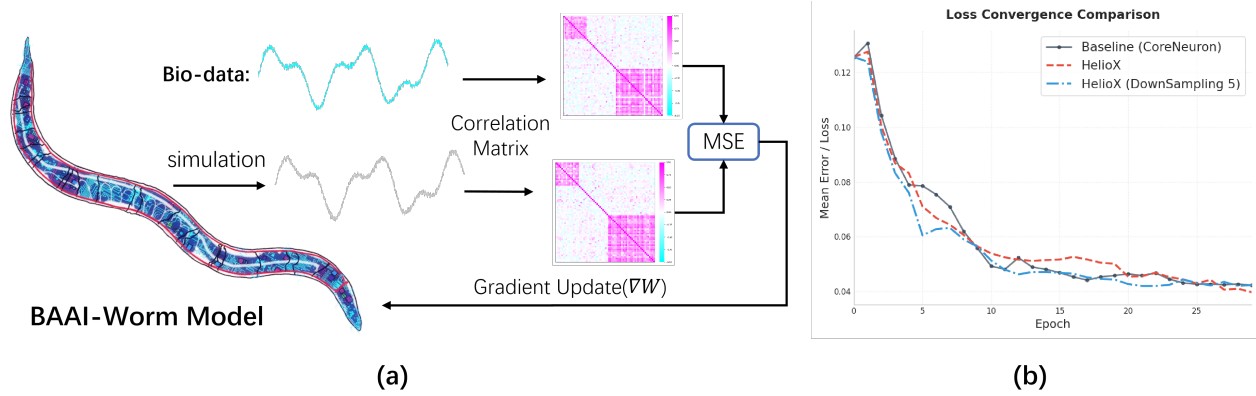

*Figure 3.* **Training the BAAIWorm whole-brain model with *HelioX*. (a)** Schematic of the biological data fitting task: optimizing chemical and electrical synaptic parameters of a 136-neuron biophysical circuit to match experimental whole-brain $Ca^{2+}$ recordings (via correlation-matrix supervision). **(b)** Training loss curves. The trajectory of *HelioX* (solid line) aligns with the CoreNEURON-based baseline (dashed line), validating numerical correctness under the same task definition.

and memory-bandwidth pressure.

We evaluate training throughput and peak memory usage on two hardware configurations: an NVIDIA RTX 4090 (48GB memory configuration), results are summarized in Table 2. With $5\times$ downsampling enabled, *HelioX* trains one epoch in about 60 seconds on an RTX 4090 while reaching the same optimization target (final loss) as the original setting, corresponding to a $\sim 247\times$ speedup over the 4.1-hour baseline.

*HelioX* substantially reduces peak memory usage ($\sim$5.2GB vs. 47GB for the baseline). These results indicate that *HelioX* significantly lowers the hardware barrier for full-brain biophysical training, enabling complex scientific tasks on standard consumer-grade GPUs.

### 5.3. Learning in Deep Biophysical Networks

Beyond the whole-brain-scale worm experiment, we also evaluate *HelioX* in a structured learning setting using a deep biophysical MLP on MNIST (LeCun et al., 2002). Following the DeepDendrite benchmark (Zhang et al., 2023), this experiment serves as a controlled framework-level study of usability, end-to-end training, and scalability across comparable learning systems.

**Network Architecture and Task Setup.** We evaluate *HelioX* on a *deep biophysical* multi-layer perceptron (MLP) for MNIST classification, following the DeepDendrite benchmark (Zhang et al., 2023) (a structured feed-forward instance of our BDN setting). The network replaces point neurons with multi-compartment Hippocampal Pyramidal Cells (HPCs): an input layer of 784 single-compartment cells (one per pixel) drives 1–3 hidden layers of 64–256 multi-compartment HPC neurons, followed by an output layer of 10 neurons.

Layers are fully connected by *graded synapses* from the presynaptic soma to a postsynaptic dendritic compartment; the trainable weight $W_{ij}$ scales the synaptic strength. For classification, we compute a voltage-based readout by averaging the somatic membrane potential $v_j(t)$ of each output neuron over a learning window $[t_s, t_e]$ (Eq. (4)), then apply a Softmax classifier and optimize cross-entropy loss.

This setup requires *HelioX* to simulate large coupled ODE systems and compute analytical gradients efficiently across the full network topology.

**Accuracy and Convergence.** We first evaluate whether *HelioX* maintains learning integrity. In a 30-epoch reproduction of the 3-layer DeepDendrite/HPC-Net MNIST setting, the *HelioX*-implemented HPC-Net reaches a test accuracy of 96.47%, compared with JAXLEY's originally reported accuracy of 94.2% in its high-fidelity setting. This parity validates that our optimization techniques do not introduce gradient drift or numerical instability. As an additional check against the original DeepDendrite/HPC-Net benchmark, *HelioX* also reproduces the same qualitative adversarial-robustness trend on MNIST: at the strongest perturbation we tested ($\epsilon = 0.20$), HPC-Net retains 86.52% accuracy, compared with 74.82% for the ANN baseline. To directly validate the analytical learning rule, we additionally compare analytical gradients against finite-difference estimates on the 3-layer biophysical MLP. Sampling 4200/52 hidden parameters and 520/10 output parameters yields cosine similarities of 0.9539 for `hidden.weight`, 0.9734 for `hidden.bias`, 0.9950 for `output.weight`, and 1.0000 for `output.bias`, showing strong directional agreement.

**Throughput and Memory.** The computational superiority of *HelioX* on this *deep biophysical MLP* task is sum-

*Table 3.* Peak GPU memory usage on the MNIST MLP task (MiB).

| Framework | Peak Memory (MiB) |
|---|---|
| CoreNEURON | 1079 |
| DeepDendrite | 1321 |
| JAXLEY | 2219 |
| *HelioX* | **1121** |

*Table 4.* Training and inference efficiency on the MLP task (time to train/test 1000 samples; batch size = 4). Rows show training (top) and testing (bottom) times in **seconds**. Entries marked N/A* are *not* missing measurements; they indicate that the corresponding public baseline implementation is fixed-topology and therefore does not support a matched depth-scaling comparison without substantial re-engineering. Architectures: **3-layer**=784-64-10, **4-layer**=784-128-64-10, **5-layer**=784-256-64-10. Input/output layers use single-compartment cells; hidden layers use PassiveHPC cells.

| Network | CoreNEURON | DeepDendrite | JAXLEY | *HelioX* |
|---|---|---|---|---|
| 3-layer | 220.99 | 9.47 | 60.41 | **5.06** |
|  | 198.57 | 10.92 | 19.24 | **4.44** |
| 4-layer | N/A* | N/A* | N/A* | **7.09** |
|  |  |  |  | **5.54** |
| 5-layer | N/A* | N/A* | N/A* | **12.32** |
|  |  |  |  | **8.65** |

marized by the throughput and memory numbers reported in Tables 4 and 3. Notably, *HelioX* outperforms JAXLEY by 11.94× on training and 4.33× on testing. Compared to DeepDendrite, *HelioX* achieves a 1.87× speedup on training and 2.46× on testing, which we attribute to our GPU-native analytical-gradient training together with multi-stream concurrency in the simulation/learning runtime. Strikingly, *HelioX* achieves a dramatic speedup of **43.67×** on training and **44.72×** on testing compared to the CoreNEURON baseline. All throughput/memory results are measured on the 3-layer network. Peak GPU memory usage is reported in Table 3. Compared to JAXLEY, *HelioX* only uses 0.51× of memory.

**Scalability.** With *HelioX*'s flexible `Sequential` API, we further evaluate scalability by training deeper *deep biophysical MLP* variants (4-layer and 5-layer); results are reported in Table 4. In contrast, the public benchmark implementations of CoreNEURON, DeepDendrite, and JAXLEY are fixed-topology scripts rather than parameterizable high-level workflows. As a result, the 4/5-layer entries for these baselines are reported as *not applicable for a matched extensibility comparison*, rather than as unmeasured runs or failed executions.

Detailed ablation results are provided in Appendix B, where we separately analyze the main simulation-side and learning-side sources of speedup.

# 6. Conclusion

We presented *HelioX*, a GPU-native framework that unifies high-performance simulation with scalable learning for biophysically detailed neural networks. *HelioX* combines a GPU-oriented simulation stack (including sparse spike processing, a unified mechanism template with a stable variable registry, multi-stream concurrency, and DHS-based tree solves) with an analytical-gradient learning runtime that avoids the overhead of generic automatic differentiation. Across both structured DBNs and an whole-brain-scale *C. elegans* circuit-fitting benchmark, our results demonstrate that *HelioX* substantially improves throughput and memory efficiency, lowering the practical barrier to training morphologically detailed networks. By co-designing a GPU-native simulator and an analytical-gradient learning runtime while remaining compatible with NEURON model specification, *HelioX* makes training and fitting of BDNs more practical across both brain-inspired AI and computational neuroscience settings.

More broadly, *HelioX* aims to make detailed neurons usable in two complementary roles: as expressive computational units for brain-inspired AI and as scalable, high-fidelity models for computational neuroscience. We hope this capability will enable both communities to iterate faster on increasingly realistic models and to explore how biological structure and dynamics shape learning and computation.

# Impact Statement

This work aims to broaden access to efficient simulation and training of biophysically detailed neural networks. Its potential positive impacts include lowering the computational and memory barriers for computational neuroscience and brain-inspired AI research, enabling more researchers to experiment with realistic neural morphologies and large-scale biophysical circuits. At the same time, faster simulation and training tools may make it easier to produce large numbers of biologically inspired models whose interpretations are not fully validated. We therefore encourage transparent reporting of modeling assumptions, careful validation of biological conclusions, and responsible use of the proposed framework.

# Acknowledgements

This work was supported by the Brain Science and Brain-like Intelligence Technology—National Science and Technology Major Project under Grant No. 2025ZD0215500/2025ZD0215502, and the National Natural Science Foundation of China (Grant No. 32471149).

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

## A. Additional Numerical Fidelity Results

**Passive 500-HPC benchmark.** For this benchmark, the most recent CoreNEURON release we tested (9.0.0) crashed when running the demo, so we report accuracy comparisons only between *HelioX* (GPU mode) and NEURON. Figure 4 shows that the observed voltage error remains within typical floating-point round-off noise.

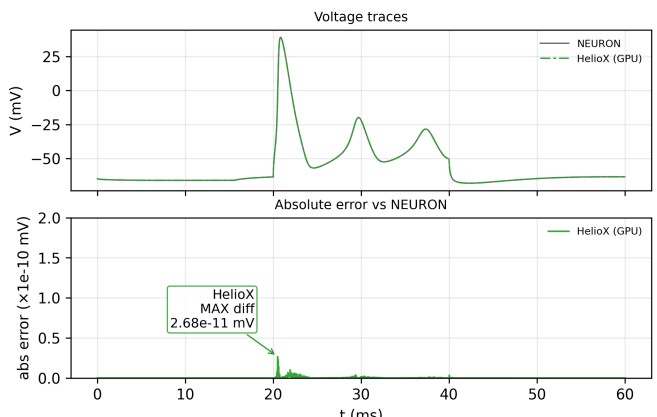

*Figure 4.* **Numerical fidelity of *HelioX* (Passive 500-HPC).** We compare *HelioX* against NEURON on the passive 500-HPC benchmark. The observed voltage error remains within typical floating-point round-off noise.

## B. Additional Ablation Results

*Table 5.* Additional ablation results. For the *C. elegans* task, percentages denote slowdown relative to the full *HelioX* pipeline under the same downsampling setting.

| Subsystem | Ablation | Slowdown |
| --- | --- | --- |
| Simulation | remove sparse spike gating | 2–3.5% |
| Simulation | serialize mechanism execution | 1.3×–2.2× |
| Simulation | disable DHS scheduling | 4.5×–10.7× |
| MNIST learning | disable GPU pixel-to-spike conversion | ∼ 3× |
| MNIST learning | disable GPU weight update/optimizer | ∼ 85× |
| Worm learning | disable GPU data access | 15% (DS=1), 375% (DS=5) |
| Worm learning | disable GPU weight update | 16% (DS=1), 366% (DS=5) |
| Worm learning | disable ring-buffer memory | 33% (DS=1), 215% (DS=5) |
| Worm learning | disable data replay | 65% (DS=1), 394% (DS=5) |

**Ablation analysis.** Table 5 isolates the main sources of speedup. On the simulation side, the dominant gain comes from GPU-native DHS scheduling, while sparse spike gating provides a smaller but still measurable benefit in the single-GPU setting. On the learning side, keeping the optimizer path on device is essential: moving weight updates off GPU induces the largest slowdown on MNIST, and the worm benchmark further shows that GPU data access, ring-buffer memory, and data replay become increasingly important when learning-time downsampling is enabled. This behavior explains why downsampling yields super-linear end-to-end speedups in practice.

## C. Preliminary Portability Evidence

Portability matters for a systems framework intended for long-term use in computational neuroscience and brain-inspired AI. Although *HelioX* is currently most mature on NVIDIA GPUs, its control logic is separated from device-specific execution through unified CPU/GPU abstractions and stable backend interfaces.

As preliminary evidence for this design, we tested an experimental ROCm backend on an AMD platform (Ryzen 7 9700X CPU, Radeon RX 7900 XT GPU) using the passive 500-HPC 1500ms simulation benchmark. Because the ROCm path is still under development, we report this result as an early portability case study rather than a fully supported benchmark setting.

*Table 6.* Preliminary portability result on the passive 500-HPC benchmark under the experimental ROCm backend. `setup_and_load_model()` includes CoreDat export, wrapper setup, and `load_model()`, and is therefore not a pure load-only timing.

| Metric | Value |
|---|---|
| Frontend build time | 3.08 s |
| NEURON simulation time | 273.49 s |
| *HelioX* `setup_and_load_model()` | 0.59 s |
| *HelioX* simulation time | 20.95 s |
| Simulation-only speedup vs. NEURON | $13.05\times$ |
| End-to-end speedup vs. NEURON | $12.70\times$ |
| Global max abs. diff. | $7.12 \times 10^{-11}$ mV |
| Global mean abs. diff. | $1.14 \times 10^{-14}$ mV |

Even in this preliminary ROCm setting, *HelioX* remains $13.05\times$ faster than NEURON in simulation and $12.70\times$ faster end-to-end, while preserving high numerical fidelity (max abs. diff. $7.12 \times 10^{-11}$ mV; mean abs. diff. $1.14 \times 10^{-14}$ mV). These results do not yet constitute a full cross-platform release, but they provide encouraging evidence that the core execution model of *HelioX* can be migrated across hardware backends while retaining both efficiency and numerical alignment.

## D. Additional Robustness Reproducibility Results

We additionally reproduce the DeepDendrite/HPC-Net-style adversarial robustness evaluation. Both MNIST and Fashion-MNIST use the full 60,000/10,000 train/test split with inputs normalized to $[0, 1]$. The target models are a *HelioX*-implemented HPC-Net with a 784-64-10 structure and a matched 784-64-10 ReLU ANN baseline. The HPC-Net checkpoints are trained for 30 epochs with batch size 4.

Adversarial examples are generated in a black-box transfer setting: an independently trained ResNet20 surrogate, adapted to $1 \times 28 \times 28$ grayscale inputs, is attacked using FGSM under an $L_\infty$ perturbation budget, and the resulting samples are evaluated on both target models. We report the full $\epsilon$ grid used in evaluation; $\epsilon = 0$ denotes the unperturbed test set. The full results are reported in Table 7.

*Table 7.* Full black-box transfer-FGSM robustness results. Attack strength $\epsilon = 0$ denotes evaluation on the unperturbed test set.

| | MNIST | | | Fashion-MNIST | |
|---|---|---|---|---|---|
| $\epsilon$ | ANN | HPC-Net | $\epsilon$ | ANN | HPC-Net |
| 0.00 | 97.66% | 96.47% | 0.00 | 88.20% | 87.29% |
| 0.02 | 97.43% | 96.30% | 0.02 | 86.44% | 86.67% |
| 0.04 | 97.18% | 96.20% | 0.04 | 84.26% | 85.93% |
| 0.06 | 96.64% | 96.09% | 0.06 | 81.41% | 85.10% |
| 0.08 | 95.78% | 95.96% | 0.08 | 78.06% | 84.34% |
| 0.10 | 94.42% | 95.83% | 0.10 | 74.23% | 83.59% |
| 0.12 | 92.15% | 95.62% | 0.12 | 70.46% | 82.54% |
| 0.14 | 88.11% | 95.17% | 0.14 | 66.10% | 80.19% |
| 0.16 | 83.61% | 93.82% | 0.16 | 61.60% | 76.55% |
| 0.18 | 79.15% | 90.62% | 0.18 | 57.61% | 71.54% |
| 0.20 | 74.82% | 86.52% | 0.20 | 53.25% | 66.97% |

These results reproduce the qualitative trend reported for DeepDendrite/HPC-Net: as adversarial strength increases, the HPC-Net architecture degrades more slowly than the ANN baseline. We include this experiment as an architecture-level reproducibility check rather than a core framework claim, since adversarial robustness is primarily task-, architecture-, and training-dependent.

