# OpenReview forum: "HelioX: A GPU-Native Framework for Simulation and Training of Biophysically Detailed Networks"
_ICML.cc/2026/Conference — ICML 2026 regular_

### Official Review · Reviewer_xM49 · 2026-03-09

**Soundness:** 4
**Presentation:** 3
**Significance:** 4
**Originality:** 3
**Overall Recommendation:** 4
**Confidence:** 3

**Summary:**

This paper introduces HelioX, a GPU-native framework designed to simulate and train biophysically detailed networks. By implementing custom CUDA kernels for the Dendritic Hierarchical Scheduling (DHS) algorithm and using analytical gradients, it successfully avoids the massive memory overhead of standard automatic differentiation. While the systems-level engineering is highly impressive, the machine learning evaluation is fundamentally inadequate.

**Compliance With Llm Reviewing Policy:**

Affirmed.

**Final Justification:**

The authors has addressed most of my concerns. So I will keep my origin score.

**Key Questions For Authors:**

My primary concern is the trivial nature of the deep learning benchmark. Detailed multi-compartment neurons possess immense power for processing spatiotemporal dynamics, yet the authors merely evaluate the framework on static MNIST classification. This fails to demonstrate the framework's actual utility for advanced AI tasks.
I will raise my score if the authors provide additional experiments on more challenging, time-dependent datasets. Specifically, I expect to see validations on event-based vision datasets (e.g., DVS-Gesture or N-Caltech101) or complex continuous-time sequence tasks. Without these, the framework's value to the broader deep learning community remains unproven.

**Limitations:**

yes

**Strengths And Weaknesses:**

Strengths:

(1) The custom CUDA implementation of DHS maximizes GPU utilization for complex, irregular dendritic structures.

(2) Massive Memory Reduction is impressive. Bypassing generic automatic differentiation drastically reduces memory overhead, enabling organism-scale training on a single GPU.

(3) The framework remains seamlessly compatible with the established NEURON modeling ecosystem.

Weaknesses:
(1) The experiment part are Toy-like ML Benchmarks.Testing a highly complex dynamic system exclusively on static MNIST images is overly simplistic and unconvincing.

(2) Relying on hard-coded analytical gradients sacrifices algorithmic flexibility, making it difficult for researchers to rapidly prototype new learning rules.

(3) The deep integration with specific CUDA primitives restricts the framework's portability across different hardware platforms. It needs more discussion on the potential application in different hardware platforms.

---

> ### Author Rebuttal · Authors · 2026-03-30
>
> We thank the reviewer for the detailed feedback and for recognizing HelioX’s contributions in GPU-native simulation, memory reduction, and NEURON compatibility. We agree that event-based vision or more complex continuous-time benchmarks would be valuable for exploring the potential of detailed neurons in temporal learning tasks.
>
> However, we want to clarify the paper’s positioning:the paper does not aim to propose a new temporal classification architecture targeting conventional deep learning benchmarks. Instead, it builds a systems framework that makes simulation and training of biophysically detailed multicompartment networks practical. More broadly, it lays the foundation for future larger-scale, higher-fidelity life simulations, such as virtual C. elegans[1] and Drosophila[2] neural models. In this sense, HelioX’s distinction from typical AI-oriented frameworks lies not in achieving top accuracy on an existing dataset, but in enabling the training of detailed biophysical neural systems to move from previously high-threshold, difficult-to-run setups to a practical research workflow.
>
> Deep Learning Benchmark
>
> Within this positioning, the MNIST experiment is an additional showcase. It demonstrates that HelioX supports structured detailed-neuron networks in end-to-end training and serves as a bridge for researchers familiar with deep learning to understand detailed neuron modeling.
>
> By contrast, the main validation is the organism-scale C. elegans whole-circuit fitting benchmark, which directly reflects the core value of detailed neuron training in bio-intelligence and computational neuroscience. This task is not a toy setting but a continuous-time, recurrent, biophysically detailed learning problem: 136 multi-compartment neurons with heterogeneous membrane mechanisms, chemical synapses, and electrical gap junctions trained under whole-brain activity constraints.
>
> Importantly, this is not static pattern recognition, but temporal learning in a complex recurrent network. Its goal is to fit biologically grounded circuit dynamics, rather than conventional classification or sequence prediction tasks. Designing networks for traditional temporal prediction is interesting but beyond the scope of this systems framework paper.
>
> Analytical Gradients Flexibility
>
> We acknowledge the limitation: compared with generic automatic differentiation, structure-aware analytical gradients are not “arbitrary learning rules plug-and-play.” This is a deliberate trade-off. For morphologically detailed neurons, generic AD across long ODE integration and large state spaces incurs high memory/computation overhead, whereas analytical gradients bypass this bottleneck.
>
> Moreover, HelioX was intentionally designed with future extensibility in mind. As presented in the main text, the framework provides a unified data-access interface, a stable variable registry, and binding mechanisms decoupled from simulator states, explicitly reserving capacity to accommodate additional objectives, training routines, and learning rules in the future.
>
> Portability
>
> We agreed that portability is important. From the outset, HelioX was designed with portability in mind: it provides unified CPU/GPU access abstractions and stable backend interfaces, decoupling control logic from hardware, leaving space for migration to other devices. An experimental ROCm backend has been preliminarily tested on a Ryzen 7 9700X CPU and RX 7900 XT GPU; in a 500-passive HPC 1200 ms simulation, HelioX achieved ~13× speedup over NEURON (273.49 s vs 20.95 s) with comparable numerical error (max abs diff 7.12e-11 mV). These results are not reported in the current paper because the ROCm backend is still under development (full training support and automatic backend switching is not yet complete).
>
> Summary
>
> We accept the reviewer’s comment regarding the limited AI-facing benchmark and will clarify in the revised manuscript that MNIST is used to showcase structured learning capability and usability, whereas the organism-scale C. elegans benchmark is the primary scientific validation, showing HelioX can move previously high-threshold, computationally expensive detailed-circuit training into a practical range. We also acknowledge that exploring detailed neurons in event-based and continuous-time tasks is valuable, but consider this a future research direction rather than a requirement for the current framework paper.
>
> References
>
> [1] Zhao, M., Wang, N., Jiang, X. et al. An integrative data-driven model simulating C. elegans brain, body and environment interactions. Nat Comput Sci 4, 978–990 (2024).
>
> [2] Shiu, P.K., Sterne, G.R., Spiller, N. et al. A Drosophila computational brain model reveals sensorimotor processing. Nature 634, 210–219 (2024).

---

> > ### Author Rebuttal · Reviewer_xM49 · 2026-04-03
> >
> > Thanks for the detailed rebuttal. It address most of my concerns. I will keep my score.

---

> > > ### Author Response · Authors · 2026-04-04
> > >
> > > We would like to sincerely thank the reviewer for the positive feedback and for acknowledging that the concerns have been fully resolved. We truly appreciate the time and effort you dedicated to reviewing our work and providing constructive suggestions, which have significantly improved the quality of this manuscript.

---

### Official Review · Reviewer_Y1bM · 2026-03-11

**Soundness:** 3
**Presentation:** 3
**Significance:** 2
**Originality:** 3
**Overall Recommendation:** 3
**Confidence:** 3

**Summary:**

Authors propose a framework to simulate and train biophysically detailed multicompartment neural networks, i.e., those that can compute spatiotemporal dynamics at the scale of C. elegans. The architecture is proposed as an improvement over the recently proposed DeepDendrite (Zhang et al., 2023) from which Dendritic Hierarchical Scheduling has been adapted.

**Compliance With Llm Reviewing Policy:**

Affirmed.

**Final Justification:**

I understand that this work has merit. But, according to the authors, the evaluations don't seem convincing for the current AI STOA. Hence, I am raising the score to 3 only. In my opinion, this looks like a good journal paper.

**Key Questions For Authors:**

Q1. How well does it perform as compared to the STOA across various metrics, as also proposed by Zhang et al. 2023 for MNIST, FMNIST datasets, etc.?

Q2. What do the features signify about the learning capabilities of the model in both the setups - deep biophysical MLPs, organism-scale C. elegans circuit-fitting benchmark?

Q3. Are these methods robust to adversarial attacks?

Q4. Why do we gain by designing HH-based deep neural networks instead of R \& F neuron-based neural networks in terms of performance on benchmarked datasets (CIFAR 100, ImageNet - you may pick other applications too that are more aligned with neuroscience)?

Q5. Since the work is also focused on training HH using backpropagation (I assume it is), how can we be sure we are indeed explaining neuroscience observations? It would be great if you could shed some light on this.

**Limitations:**

Yes

**Strengths And Weaknesses:**

**Strengths**
1. This work is scientifically sound, and the design choices are well explained.
1. Memory and Latency gains over prior methods, enabling scaling up to biophysically-detailed networks.

**Weakness**
1. Insufficient Experiments: The paper showcases performance for toy MLP baselines with incomplete experiments in the tables (Table 1 and Table 3).
2. No STOA accuracy comparison (Zhang et al., 2023).
3. No ablations on the efficiency of the training method.
4. Zhang et al. (2023 show robustness to adversarial attacks in their HPC-Net architecture using DeepDendrite. Authors don't discuss any such capabilities in their networks.
5. If the goal is to explain neuroscience observations, then training with biologically implausible backpropagation is not well motivated.
6. If the goal is brain-inspired AI, then some rigorous experiments on large-scale datasets (ImageNet, CIFAR 100) are necessary. The authors should also highlight the advantage of HH models over computationally simpler Resonate and Fire-based neural networks.

---

> ### Author Rebuttal · Authors · 2026-03-29
>
> We thank the reviewer for the detailed feedback. We believe part of the disagreement stems from the positioning of our work. HelioX is not proposed as a new HH-based architecture for standard AI benchmarks, nor does it aim to compete with conventional deep learning models on ImageNet/CIFAR-100. Instead, its core contribution is a systems framework that makes simulation and gradient-based training of biophysically detailed multicompartment networks practical. In this context, the MLP experiments are a controlled showcase, while the organism-scale C. elegans circuit-fitting benchmark is our primary scientific validation.
>
> Q1. Comparison with DeepDendrite
>
> On MNIST, after 30 epochs, HelioX achieves 96.47% accuracy, compared to 96.46% reported by DeepDendrite (Zhang et al., 2023). On FMNIST, HelioX achieves 87.29%, compared to 86.04% in the original DeepDendrite paper. These results show that, within the same class of detailed-neuron learning benchmarks, HelioX reaches performance comparable to prior work.
>
> Our point, however, is not to introduce a new task-specific learning algorithm or architecture for higher benchmark accuracy. Rather, the MLP experiments show that, under the same learning paradigm and model family, HelioX supports more efficient training and more natural structural scaling.
>
> Q2. Role of the two experimental setups
>
> The MLP setup shows that detailed neuron models can support learning tasks and that HelioX enables end-to-end training of structured multi-layer detailed networks with a usable high-level API. The C. elegans whole-circuit fitting benchmark evaluates whether the framework can scale to biologically detailed recurrent circuits; importantly, it is not merely “larger,” but a behaviorally meaningful, biologically grounded detailed-circuit learning setting derived from BAAIWorm.
>
> Q3. Robustness
>
> Yes. We additionally evaluated robustness using a black-box transfer-FGSM setting: a ResNet20 surrogate was trained to generate adversarial examples, and both ANN and HPC-Net were evaluated on the same perturbed inputs.
>
> We observe the same qualitative trend on both MNIST and FMNIST: as attack strength increases, HPC-Net exhibits smaller degradation than ANN. For example, on MNIST at ϵ=0.20, ANN drops from 97.66% to 74.82%, while HPC drops from 96.47% to 86.52%. On FMNIST at ϵ=0.20, ANN drops from 88.20% to 53.25%, while HPC drops from 87.29% to 66.97%.
>
> This shows that HelioX supports robustness analyses consistent with prior work and reproduces the same qualitative HPC-Net trend. Robustness is important, but it is strongly task-, architecture-, and training-dependent, and is not the central claim of this framework paper.
>
> Q4. Why HH-based detailed neurons instead of simpler R&F models?
>
> We do not claim that HH-based multicompartment networks are the optimal choice for standard image classification benchmarks. If the goal is only accuracy and throughput on conventional vision tasks, simpler neuron models are often more practical.
>
> Our motivation is not benchmark accuracy, but that HH-based detailed neurons preserve morphology, multi-compartment membrane dynamics, dendritic computation, and experimentally grounded biophysical processes. These properties are necessary for biophysical fidelity, mechanistic analysis, and organism-scale detailed-circuit training. HelioX is designed for scenarios where biological realism and circuit-level modeling matter, rather than as a replacement for standard deep learning frameworks.
>
> Q5. Biological interpretation and backpropagation
>
> We do not claim that biological brains literally implement backpropagation. Here, gradient-based optimization is better viewed as a parameter-fitting / system-identification tool for fitting a biologically detailed forward model. Its scientific value comes from the model itself, which retains morphology, membrane mechanisms, synapses, and circuit dynamics, and can therefore be compared against experimental observations.
>
> In the C. elegans setting, this benchmark is derived from the BAAIWorm work (Nature Computational Science, 2024), which constructs a closed-loop brain–body–environment model and reproduces realistic zigzag locomotion toward attractors. Thus, its significance is not only scale or computational cost, but also its connection to behaviorally meaningful biological phenomena. HelioX does not redefine this scientific problem; rather, it makes this class of biologically grounded detailed-circuit training practically accessible.
>
> Final remark
>
> Our main claim is not SOTA accuracy on standard deep learning benchmarks, but that HelioX makes organism-scale detailed biophysical training far more practical and reproducible.
>
> Appendix: Adversarial Robustness Results
>
> Attack strength|ANN|HPC
>
> FMNIST:
>
> 0.00|88.20|87.29
>
> 0.04|84.26|85.93
>
> 0.08|78.06|84.34
>
> 0.12|70.46|82.54
>
> 0.16|61.60|76.55
>
> 0.20|53.25|66.97
>
> MNIST
>
> 0.00|97.66|96.47
>
> 0.04|97.18|96.20
>
> 0.08|95.78|95.96
>
> 0.12|92.15|95.62
>
> 0.16|83.61|93.82
>
> 0.20|74.82|86.50

---

> > ### Author Rebuttal · Reviewer_Y1bM · 2026-04-03
> >
> > Thanks for the clarification! I understand that this work has merit. But, according to the authors, the evaluations don't seem convincing for the current AI STOA. Hence, I am raising the score to 3.

---

> > > ### Author Response · Authors · 2026-04-07
> > >
> > > We appreciate the reviewer’s concern and agree that this is an important question. In the current paper, we do not evaluate HelioX on large-scale benchmark datasets or event-driven application datasets, and therefore we do not claim that the practical value of multi-compartment biophysical networks for such tasks has already been established.
> > >
> > > We also agree that, before strong results are demonstrated on truly large-scale datasets/tasks, the broader application significance of multi-compartment models cannot be considered fully settled. However, our goal in this paper is not to resolve that question in one step. Rather, HelioX is intended as a preceding and enabling step: its purpose is to substantially reduce the computational and memory cost of gradient-based simulation and training for biophysically detailed networks, which has been a major bottleneck in this line of work. This framing is also consistent with the paper’s main contribution as a GPU-native framework for scalable simulation and training of biophysically detailed multi-compartment networks.
> > >
> > > As for the practical utility of BDNs on large-scale datasets, we agree that this remains to be validated. To the best of our knowledge, prior work has not yet demonstrated gradient-based training of multi-compartment biophysical networks at the scale of datasets such as ImageNet. **In this sense, we view HelioX not as the final answer to that question, but as an infrastructure step that substantially lowers the barrier and moves the field closer to making such experiments feasible.** We agree that the types of tasks you mention are important, and they are precisely the kinds of follow-up studies that HelioX is designed to enable.

---

### Official Review · Reviewer_LVjR · 2026-03-12

**Soundness:** 3
**Presentation:** 3
**Significance:** 2
**Originality:** 2
**Overall Recommendation:** 4
**Confidence:** 3

**Summary:**

This paper introduces HelioX, a GPU-native framework for simulating and training morphologically detailed biophysical neural networks. The system combines a GPU implementation of dendritic tree solvers, sparse spike processing, and an analytical-gradient training module to avoid the overhead of generic autodiff frameworks. Experiments show improved runtime and memory efficiency compared to existing simulators and demonstrate training of biophysical networks on tasks such as MNIST classification.
Overall, the paper aims to make large-scale training of biophysically detailed neuron models more practical.

**Compliance With Llm Reviewing Policy:**

Affirmed.

**Final Justification:**

The paper presents a GPU-native framework for simulating and training biophysically detailed neural networks, addressing an important problem at the intersection of ML systems and computational neuroscience, and demonstrating compelling improvements in runtime and memory.
In my original review, I raised concerns about (1) unclear attribution of speedups, (2) limited validation of the analytical gradients, and (3) lack of clarity in comparisons with JAXLEY. The author rebuttal has addressed these points.
While some comparisons remain imperfect, I am satisfied that the core technical contributions are sound. The novelty lies primarily in system integration and engineering, but the resulting framework appears practically useful.
Overall, the rebuttal positively changed my assessment, and I weakly support acceptance.

**Key Questions For Authors:**

1. Can the authors provide ablations isolating the contributions of the main system components?
2. What configuration exactly caused JAXLEY to fail in the deeper network experiments?

**Limitations:**

yes

**Strengths And Weaknesses:**

Strengths:
- Important problem.

- The GPU-native design and engineering effort appear substantial.

- The reported improvements in runtime and memory are promising and could enable larger experiments.

- Realistic tasks. The experiments include end-to-end learning workloads.

Weaknesses:
- Unclear source of speedups. The paper combines several design choices but it is not clear which components drive the performance gains. Ablations isolating the contributions of the GPU solver, analytical gradients, and scheduling optimizations would make the systems contribution clearer.

- Limited validation of the analytical gradient formulation. The paper demonstrates that the forward simulation matches existing simulators and shows that networks trained with HelioX achieve good performance on tasks such as MNIST. This provides indirect evidence that the gradients are useful for training. However, additional validation (e.g., gradient checks or comparisons with autodiff-based implementations on smaller models) would strengthen confidence that the analytical gradients faithfully capture the underlying simulator dynamics.

- Comparison with JAXLEY needs clarification. The paper reports that JAXLEY cannot run certain deeper network settings, but the JAXLEY paper demonstrates large-scale training of biophysical networks. The exact configuration used (e.g., simulation length, checkpointing, hardware limits) should be clarified. Additionally, several entries for JAXLEY in the comparison tables are missing rather than measured; it would be helpful to explain whether these results could be obtained or why they were omitted.

- Novelty mostly in system integration. Some core components are taken directly from prior work (e.g., DHS, analytical gradient), suggesting that the main contribution lies primarily in system engineering and integration rather than fundamentally new algorithms.

---

> ### Author Rebuttal · Authors · 2026-03-28
>
> We thank the reviewer for the insightful comments on ablation, gradient validation, cross-framework comparison, and novelty. We agree these are important for clarifying our contributions.
>
> HelioX is not intended as a general-purpose MLP framework, but as a systems framework that makes high-fidelity simulation and training of biophysically detailed networks practical. MNIST serves as a controlled benchmark for usability and end-to-end training, while the C. elegans whole-circuit fitting task is our primary validation. This task is not a toy setting, but an organism-scale, recurrent, biophysically detailed learning problem, and represents a meaningful frontier benchmark in this domain.
>
> Q1. Speedup Source
>
> Simulation-side ablations.
>
> removing sparse spike gating: 2–3.5% slowdown
> serializing mechanism execution: 1.3×–2.2× slowdown
> disabling DHS solver scheduling: 4.5×–10.7× slowdown
>
> This shows the primary gains come from GPU-native DHS solving and concurrent execution, while spike optimization is secondary under a single-GPU setting.
>
> Learning-side ablations
>
> MLP:
>
> Disabling GPU pixel-to-spike conversion: ~3× slowdown
>
> Disabling GPU weight update & optimizer interface: ~85× slowdown
>
> C. elegans (200 ms window):
>
> Disabling GPU data access: 15% slowdown (no DS), 375% with 5× DS
>
> Disabling GPU weight update: 16% (no DS), 366% with 5× DS
>
> Disabling ring-buffer memory: 33% (no DS), 215% with 5× DS
>
> Disabling data replay: 65% (no DS), 394% with 5× DS
>
> Notably, learning-time downsampling leads to super-linear speedup. Some optimizations reflect system-level co-design, including GPU memory layout and data access; removing them is roughly equivalent to rewriting the framework and cannot be independently ablated.
>
> Q2. Gradient Validation
>
> We compare analytical gradients with finite-difference gradients on a 3-layer biophysical MLP (sampling 4200/52 hidden params and 520/10 output params). Cosine similarities:
>
> hidden.weight: 0.953871/hidden.bias: 0.973361/output.weight: 0.994982/output.bias: 0.999976
>
> This demonstrates strong directional agreement, supporting the correctness of our analytical gradients.
>
> Q3. Missing JAXLEY Results
>
> Simulation.
> The key issue is not whether models can run, but whether they can be compared under strict NEURON-aligned matched settings. NEURON is the de facto reference simulator in this domain[1], and alignment is necessary to ensure numerical correctness and scientific validity of comparisons.
>
> We encountered three main challenges when aligning JAXLEY:
>
> 1.morphology/discretization mismatch
>
> 2.mechanism alignment differences
>
> 3.lack of delayed event-driven spike handling
>
> Even after careful alignment, JAXLEY shows non-negligible discrepancies (0.0817 mV without spikes, up to 12.89 mV with spikes).
> Thus, we report only strictly matched settings and will clarify configurations in the final version.
>
> Learning.
> The 4/5-layer entries are not meant to imply other frameworks cannot scale. Instead, their available benchmark implementations are fixed-topology scripts rather than parameterizable high-level workflows.
>
> Extending them would require non-trivial restructuring of their implementation paradigm, which would no longer constitute a direct matched comparison. Therefore, we do not include these entries.
>
> Q4. Novelty
>
> We partially agree that system integration is central, but HelioX is not merely integration—it combines two independently strong subsystems:
>
> Simulation.
> HelioX achieves 1.61–2.25× speedup over CoreNEURON, 1.69–2.60× over DeepDendrite, and 3.12× over JAXLEY, while maintaining <1e-8 mV numerical error and identical spike timing.
> Importantly, building a high-performance simulator that remains numerically aligned with NEURON is itself a recognized systems contribution in this field[2], and HelioX already reaches this level.
>
> Learning.
> HelioX achieves 11.94× training speedup over JAXLEY and 1.87× over DeepDendrite on MLP, and reduces C. elegans training memory from 47.4 GB to 5.2 GB (1.6 GB with downsampling), with epoch time reduced to ~60 s.
> While we avoid claiming universal SOTA due to unmatched settings, HelioX is competitive with or superior to strong baselines in reported settings.
>
> Integration.
> The key contribution is that these two strong components are unified in a NEURON-compatible GPU-native framework.
>
> More importantly, this integration breaks a practical barrier: it brings an organism-scale detailed-circuit training problem—previously requiring extremely high computational cost—into a regime feasible on a single consumer GPU.
>
> Therefore, the contribution is not “only system integration,” but the first unification of high-performance simulation and efficient learning into a practical detailed-biophysical workflow, substantially lowering the barrier for this class of research.
>
> References
>
> [1] ModelDB: https://modeldb.science/FindBySimulator
>
> [2] Kumbhar et al., CoreNEURON: An Optimized Compute Engine for the NEURON Simulator, Front. Neuroinform., 2019.

---

> > ### Author Rebuttal · Reviewer_LVjR · 2026-04-04
> >
> > I thank the authors for their rebuttal. The ablation results and gradient cosine similarity checks have adequately addressed my key concerns. I would like to ask the authors to ensure that these additions are incorporated into the final manuscript. Including the ablation analysis, gradient validation results, and clarifications on the JAXLEY comparison.

---

> > > ### Author Response · Authors · 2026-04-04
> > >
> > > We sincerely thank the reviewer for the positive feedback and for acknowledging that our rebuttal and the additional experiments have addressed the concerns. Following your suggestion, we confirm that all these additions—including the ablation analysis, gradient validation results, and clarifications regarding the JAXLEY comparison—will be fully incorporated into the final manuscript. We believe these inclusions will significantly strengthen the clarity and rigor of our work. Thank you again for your constructive guidance throughout the review process.

---

### Official Review · Reviewer_fyqn · 2026-03-13

**Soundness:** 3
**Presentation:** 2
**Significance:** 3
**Originality:** 3
**Overall Recommendation:** 4
**Confidence:** 2

**Summary:**

The paper addresses the inefficiency of general-purpose deep learning frameworks in handling biophysically detailed neural networks (BDNs). It proposes HelioX, a GPU-native framework that incorporates optimizations such as DHS-based parallel tree solving and multi-stream concurrency. Experimental results demonstrate that the proposed framework significantly improves computational efficiency and scalability compared with existing general-purpose deep learning frameworks.

**Compliance With Llm Reviewing Policy:**

Affirmed.

**Final Justification:**

The authors clarify that HelioX is, in principle, capable of supporting the training of various biophysical parameters; however, the practical flexibility and generality of the learning module remain only partially validated, as empirical results on more complex parameter learning scenarios (e.g., ion channel dynamics optimization) are still lacking. Therefore, I maintain my current rating.

**Key Questions For Authors:**

The main questions for the authors are described in the Weaknesses section above.

**Limitations:**

Yes

**Strengths And Weaknesses:**

Strengths:
1. The paper addresses the inefficiency of general-purpose deep learning frameworks in handling BDNs and proposes HelioX, a GPU-native framework. The method presents a systematic system-level design, incorporating mechanisms such as DHS-based parallel tree solving and multi-stream concurrent execution to better exploit GPU parallelism for tree-structured neuronal computations. Overall, the framework is clearly structured and well-tailored to the characteristics of the target problem.
2. The authors conduct extensive experiments across multiple simulation and training scenarios. The results show consistent improvements in computational efficiency, including significant speedups over existing frameworks such as CoreNEURON and substantial reductions in memory consumption, supporting the effectiveness of the proposed system.

Weaknesses:
1. The paper explicitly states that HelioX relies on NEURON for model construction, such as morphology parsing and mechanism insertion, using NEURON as a front-end modeling tool while HelioX serves as the GPU backend for execution. This coupling may limit the framework’s independent extensibility; updates or interface changes in NEURON could potentially affect HelioX’s compatibility. In addition, users are required to be familiar with both NEURON and HelioX workflows, which may increase the learning cost.
2. Although HelioX adopts analytic gradient methods to avoid the high computational and memory overhead associated with automatic differentiation, the learning mechanisms demonstrated in the paper mainly focus on synaptic weight updates based on analytic synaptic gradients (e.g., those used in DeepDendrite). For more complex biophysical parameter learning problems—such as ion channel dynamics optimization or dendritic spine plasticity—the flexibility and generality of the framework’s learning module remain to be further validated.
3. Although the paper demonstrates efficiency improvements across several simulation and training tasks, a systematic evaluation of scalability on larger neural network models—such as more complex biological nervous systems or higher-resolution neuromorphological structures—is still lacking. Further analysis of performance across different hardware scales or more complex network architectures would help provide a more comprehensive assessment of the framework’s scalability.

---

> ### Author Rebuttal · Authors · 2026-03-28
>
> We thank the Reviewer for their thoughtful and constructive feedback. We are encouraged that they recognize HelioX's system-level value and share our vision for **Brain-Inspired Intelligence**, which promises more efficient, general-purpose AI. Driven by breakthroughs like the BAAI Worm[2] and the Drosophila connectome[3], the field is rapidly approaching organism-level whole-brain modeling and simulation. **HelioX** serves as a *high-performance catalyst* for this transition, providing the computational foundation to model, simulate, and train increasingly complex, biologically detailed neural circuits at scale.
>
> Regarding the specific concerns raised, we acknowledge that certain aspects of our initial description could be further clarified. We appreciate the opportunity to resolve any ambiguity and provide the following detailed clarifications.
>
> ## Q1: NEURON coupling
>
> Regarding the reviewer’s concern about coupling between NEURON and HelioX, we note that NEURON is the de facto standard in detailed neuron modeling. In the open-source ModelDB repository, the majority of models are implemented in NEURON, surpassing any dedicated frameworks or general-purpose languages (Python, C++, etc.). Excluding models implemented in general-purpose languages, NEURON-based models account for approximately 58% of all models in ModelDB. Therefore, NEURON compatibility is critical for rapidly transferring decades of accumulated neuron model ecosystems. This is why our paper emphasizes HelioX’s compatibility and numerical consistency with NEURON—something other works cannot easily achieve, and one of the highlights of our system.
>
> Regarding the concern that HelioX may be “over-coupled” with NEURON, potentially limiting future extensibility, we considered this from the project’s inception. HelioX’s dependency on NEURON is limited to the exported model description (which is relatively stable). After model construction, HelioX imports the model into its GPU-native runtime, where trainable parameters are bound to stable runtime identifiers. The learning backend operates directly on the simulator states on the device, decoupling the high-frequency training loop from frontend object semantics. We will clarify this design intention more explicitly in the revised manuscript.
>
> Additionally, HelioX provides a high-level Sequential API for structured biophysical networks, allowing researchers unfamiliar with NEURON to use the framework conveniently. The current runtime design also leaves room for alternative frontends.
>
> ## Q2: Flexibility of learning modules
>
> We acknowledge the reviewer’s observation. In detailed neuron simulation, networks can be divided into two parts: (1) the morphological skeleton and (2) the membrane mechanisms (“Mech”). In the simulator implementation, ion channels, synapses, etc., are represented as mechanisms with unified state-update and parameter management interfaces, and in principle, all are trainable in HelioX.
>
> In the original BAAIWorm setup, many ion channel parameters come from experimental data. To ensure scientific comparability with prior work, we retain the same modeling choices, focusing optimization on chemical synapses and electrical gap junctions rather than extensively refitting all membrane mechanism parameters. This ensures fair comparisons with prior work. We will make this point more explicit in the final manuscript.
>
> ## Q3: Scalability
>
> We agree that larger nervous systems are an important future direction. However, even before HelioX, tasks at the scale of BAAIWorm[2] were already extremely demanding. In our paper, the CoreNEURON baseline requires 47.4 GB of peak GPU memory for this task, indicating that scaling further is practically difficult. Our contribution is to push this frontier forward: HelioX reduces peak memory to 5.2 GB, and down to 1.6 GB with learning-time downsampling; epoch time drops from 14,840 s to 5,100 s, and with 5× downsampling to 60 s on a single RTX 4090.
>
> Thus, the C. elegans whole-circuit setup represents one of the most challenging organism-scale detailed-circuit training scenarios known. HelioX significantly lowers the hardware barrier, making such training feasible for the first time. While scaling to larger models (e.g., zebrafish or Drosophila) is an exciting future direction, it exceeds the scope of this systems framework paper. The core contribution here is advancing a previously practically inaccessible frontier to actual usability.
>
> ## References
> [1] ModelDB: https://modeldb.science/FindBySimulator, accessed March 27, 2026.
>
> [2] Zhao, M., Wang, N., Jiang, X. et al. An integrative data-driven model simulating C. elegans brain, body and environment interactions. Nat Comput Sci 4, 978–990 (2024). https://doi.org/10.1038/s43588-024-00738-w
>
> [3] Shiu, P.K., Sterne, G.R., Spiller, N. et al. A Drosophila computational brain model reveals sensorimotor processing. Nature 634, 210–219 (2024). https://doi.org/10.1038/s41586-024-07763-9

---

> > ### Author Rebuttal · Reviewer_fyqn · 2026-04-02
> >
> > I appreciate the authors’ response, but I still have some concerns. The second reply indicates that HelioX is, in principle, capable of supporting the training of a variety of biophysical parameters. However, the practical flexibility and generality of the learning module are only partially addressed, as empirical validation on more complex parameter learning scenarios (e.g., ion channel dynamics optimization) has not yet been provided. Therefore, I am inclined to maintain my current rating.

---

> > > ### Author Response · Authors · 2026-04-07
> > >
> > > We appreciate the reviewer’s follow-up and agree that this is an important point. In our previous rebuttal, we did not yet include a dedicated empirical validation for ion-channel parameter learning. To directly verify that the HelioX learning module also supports non-synaptic biophysical parameter optimization, we have added additional ion-parameter fitting experiments.
> > >
> > > ## Cross-mechanism ion-channel parameter learning
> > >
> > > We first conducted a cross-mechanism ion-parameter fitting experiment on a single L5PC neuron, where two ion-channel parameters from different mechanisms were optimized jointly: **axon_gnabar_hh3_la** (from **hh3_la**) and **tuft_ghdbar_hd_la** (from **hd_la**). The teacher target was generated by forward simulation with fixed ground-truth parameters under four stimulation settings (**near_0495**, **strong_0500**, **strong_0510**, and **strong_0530**). To create a clear initial mismatch, both trainable parameters were initialized at **0.8×** of their target values. The loss was the same differentiable **soft_hybrid** objective used in our previous experiments, combining trajectory-matching and spike-related terms. Gradients were computed through our analytical pipeline, i.e., mechanism-internal **di/dθ** together with replay-based transfer-resistance backpropagation, rather than generic automatic differentiation.
> > >
> > > On the **teacher_axon_clean** task, this experiment showed stable and strong convergence: the loss decreased from **18.2629** initially to **8.44e-06** at the best epoch (epoch 13), and remained at **9.83e-05** at epoch 20. The spike responses under the four stimulation settings also converged to the teacher behavior, changing from **(0, 0, 0, 1)** before training to **(0, 1, 1, 4)** after convergence.
> > > which matches the teacher neuron. The learned parameter values also returned to a biologically reasonable range:
> > >
> > > * **axon_gnabar_hh3_la:** 2.40 → 2.99
> > > * **tuft_ghdbar_hd_la:** 0.0008 → 0.001057
> > >
> > > These results provide direct empirical evidence that the HelioX learning module is not limited to synaptic-weight optimization, but can also support joint optimization of non-synaptic ion-channel parameters across different mechanisms.
> > >
> > > ## Follow-up harder setting: opposite-direction initialization
> > >
> > > We further considered that the above **0.8× / 0.8×** initialization might still be relatively favorable, since both parameters start from the same side of the target. To test a more challenging case, we performed a follow-up experiment with **opposite-direction mismatch initialization**. The teacher target remained the same, with spike responses **(0, 1, 1, 4)** under the four stimulation protocols. We then initialized the two trainable parameters on opposite sides of the target: the axonal fast Na conductance parameter was set to **1.2×** of its target value, while the tuft Ih-related conductance parameter was set to **0.8×**. Given target values of **3.0** and **0.001**, the corresponding initial values were **3.6** and **0.0008**, respectively. This setting was trained directly for **20 epochs**.
> > >
> > > ### Results of the harder setting
> > >
> > > This opposite-direction initialization is substantially more challenging. The model starts from a strongly over-spiking regime (1, 4, 4, 7) with loss 27.1687, reaches the correct teacher spike pattern (0, 1, 1, 4) at its best point with loss 0.0054587, but partially falls out of that regime by the end of training (0, 0, 1, 1) with final loss 12.0801. This suggests that cross-mechanism ion-parameter fitting is a genuinely nontrivial optimization problem with threshold/basin-switching behavior, rather than an overly easy setting.
> > >
> > > In other words, this harder initialization exhibits a **“first entering the correct firing basin, then falling out again”** phenomenon. We interpret this as evidence that cross-mechanism ion-parameter fitting is a genuinely nontrivial optimization problem, especially when different parameters are initialized on opposite sides of the target and jointly affect spike thresholds and trajectory-level responses.
> > >
> > > ## Overall implication
> > >
> > > Taken together, these experiments strengthen our response to the reviewer’s concern in two ways.
> > >
> > > First, they provide direct empirical evidence that the HelioX learning module is not restricted to synaptic gradients, but can also optimize **ion-channel-related biophysical parameters**, including **joint training of parameters from different mechanisms**.
> > >
> > > Second, the harder opposite-direction initialization shows that this capability is not limited to an overly easy setting: **HelioX** can still move the neuron toward the correct teacher behavior even in a more difficult optimization landscape, although the experiment also reveals that such cross-mechanism ion-parameter learning can involve unstable threshold-basin transitions. We believe this is precisely the kind of empirical evidence needed to support the practical flexibility of the HelioX learning module beyond synaptic-weight optimization.

---

### Decision · Program_Chairs · 2026-04-30

**Decision:**

Accept (regular)

**Comment:**

**Summary of reviews.** Four reviewers: three Weak Accept (4), one Weak Reject (3, raised from an apparent initial score of 2). The paper presents a GPU-native framework for simulating and training biophysically detailed (multi-compartment, ion-channel-resolved) neural networks. Custom fused CUDA kernels implement Dendritic Hierarchical Scheduling (DHS) for irregular dendritic tree structures, and analytical gradient computation replaces generic autodiff, enabling organism-scale training on a single consumer GPU. The C. elegans whole-circuit (BAAIWorm-scale) fitting is demonstrated.

**Key strengths.**
- Addresses an important infrastructure gap: no existing framework can efficiently simulate AND train biophysically detailed networks on GPU at organism scale (consensus).
- Substantial engineering contribution: DHS scheduling, analytical gradients, multi-stream concurrency achieve orders-of-magnitude speedup over CoreNEURON and significant memory reduction (consensus).
- Seamless compatibility with the established NEURON ecosystem (Reviewer xM49).
- Gradient validation: cosine similarity 0.954–1.000 vs. finite-difference gradients (Reviewer LVjR satisfied after rebuttal).
- Ablation data provided during rebuttal: DHS removal causes 4.5–10.7× slowdown; GPU weight update removal causes ~85× slowdown (Reviewer LVjR).

**Key weaknesses.**
- **ML benchmarks are toy-level** (MNIST, small MLPs) — all four reviewers flagged this. The authors' defense is that this is a systems/infrastructure framework, not a classification architecture, and that C. elegans circuit fitting is the primary scientific validation. This framing is reasonable but makes the paper harder to evaluate at a standard ML venue.
- **Learning generality partially unvalidated** (Reviewer fyqn): initial validation was only for synaptic weight learning. Authors added ion-channel parameter fitting experiments during rebuttal, but Reviewer fyqn found this only "partially" convincing.
- **Novelty lies primarily in system integration** rather than fundamentally new algorithms (Reviewer LVjR) — DHS and analytical gradients build on prior work. However, the engineering contribution of making them work together at scale on GPU is non-trivial.
- **Hardware portability limited** by deep CUDA integration (Reviewer xM49) — partial AMD ROCm support shown as preliminary.
- Reviewer Y1bM (the holdout at 3) argues the paper is "better suited as a journal paper" and that evaluations are not convincing for AI state-of-the-art. This is a valid venue-fit concern.